# Inhaled ciclesonide in adults hospitalised with COVID-19: a randomised controlled open-label trial (HALT COVID-19)

Daniel Brodin,[1] Per Tornhammar [ID],[2] Peter Ueda,[3] Anders Krifors,[4,5] Eli Westerlund [ID],[6] Simon Athlin,[7] Sandra Wojt,[8] Olof Elvstam,[9] Anca Neumann,[1] Arsim Elshani,[10] Julia Giesecke,[2] Jens Edvardsson-Källkvist,[11] Sayam Bunpuckdee,[2] Christian Unge,[8] Martin Larsson,[12,13] Björn Johansson,[14] Johan Ljungberg,[14] Jonas Lindell,[15] Johan Hansson,[16] Ola Blennow,[1,17] Daniel Peter Andersson [ID] [18]

DB, PT and PU contributed equally.
OB and DPA contributed equally.

For numbered affiliations see end of article.

**Correspondence to**
Dr Ola Blennow;
ola.blennow@regionstockholm.se

## ABSTRACT

**Objective** To assess the efficacy of inhaled ciclesonide in reducing the duration of oxygen therapy (an indicator of time to clinical improvement) among adults hospitalised with COVID-19.

**Design** Multicentre, randomised, controlled, open-label trial.

**Setting** 9 hospitals (3 academic hospitals and 6 non-academic hospitals) in Sweden between 1 June 2020 and 17 May 2021.

**Participants** Adults hospitalised with COVID-19 and receiving oxygen therapy.

**Intervention** Inhaled ciclesonide 320 µg two times a day for 14 days versus standard care.

**Main outcome measures** Primary outcome was duration of oxygen therapy, an indicator of time to clinical improvement. Key secondary outcome was a composite of invasive mechanical ventilation/death.

**Results** Data from 98 participants were analysed (48 receiving ciclesonide and 50 receiving standard care; median (IQR) age, 59.5 (49–67) years; 67 (68%) men). Median (IQR) duration of oxygen therapy was 5.5 (3–9) days in the ciclesonide group and 4 (2–7) days in the standard care group (HR for termination of oxygen therapy 0.73 (95% CI 0.47 to 1.11), with the upper 95% CI being compatible with a 10% relative reduction in oxygen therapy duration, corresponding to a <1 day absolute reduction in a post-hoc calculation). Three participants in each group died/received invasive mechanical ventilation (HR 0.90 (95% CI 0.15 to 5.32)). The trial was discontinued early due to slow enrolment.

**Conclusions** In patients hospitalised with COVID-19 receiving oxygen therapy, this trial ruled out, with 0.95 confidence, a treatment effect of ciclesonide corresponding to more than a 1 day reduction in duration of oxygen therapy. Ciclesonide is unlikely to improve this outcome meaningfully.

**Trial registration number** NCT04381364.

## STRENGTHS AND LIMITATIONS OF THIS STUDY

⇒ This was a multicentre, randomised, controlled, open-label trial comparing treatment with the inhaled corticosteroid ciclesonide 320 µg two times a day for 14 days versus standard care.

⇒ Healthcare providers and participants were not blinded to treatment assignment.

⇒ The trial was terminated early due to slow recruitment.

## INTRODUCTION

Patients with COVID-19 can develop acute respiratory failure that may require invasive mechanical ventilation, associated with high mortality. The unregulated inflammation in the lungs, poor oxygenation and pulmonary infiltrates characterising severe COVID-19 have been considered as a type of acute respiratory distress syndrome (ARDS).[1 2]

Prior to the COVID-19 pandemic, studies have indicated that inhaled corticosteroids may reduce the risk of ARDS. In a randomised controlled trial including 61 patients at risk of ARDS, none of the patients assigned to aerosolised budesonide/formoterol versus 7 assigned to placebo developed ARDS,[3] and 6 (20%) and 16 (53%) of the patients, respectively, received mechanical ventilation. In another trial including 60 patients with acute lung injury or ARDS, nebulised budesonide improved oxygenation and peak and plateau airway pressures, and reduced inflammatory markers.[4] Moreover, potentially protective and preventive effects of inhaled corticosteroids for ARDS are supported by animals models of lung injury,[5–8] and in vitro studies,[9] and it has been speculated that local

administration of the drug in the lung may maximise therapeutic benefits with fewer systemic side effects, as compared with systemic steroids.[3]

Therefore, it could be hypothesised that inhaled corticosteroids may be beneficial for patients with severe COVID-19. The hypothesis is further supported by reports that inhaled corticosteroids reduce the epithelial expression of genes linked to SARS-CoV-2 entry into host cells.[10 11] Among the inhaled corticosteroids, ciclesonide has been identified as a particularly promising treatment as it can suppress replication of SARS-CoV-2 in vitro.[12 13]

While previous randomised controlled trials have assessed the effects of inhaled budesonide[14 15] or ciclesonide[16 17] in patients non-hospitalised with COVID-19, no study has been performed in hospitalised patients with more severe COVID-19.

This open-label randomised controlled trial investigated the effects of inhaled ciclesonide, compared with standard care, in adult patients hospitalised with COVID-19 and requiring oxygen therapy.

## METHODS
### Study design
The HALT COVID-19 (inHALation of cliclesonide for Treatment of COVID-19) trial was a multicentre, open-label randomised controlled trial to assess the efficacy and safety of inhaled ciclesonide for the treatment of patients hospitalised with COVID-19 receiving oxygen therapy. The trial was conducted at nine hospitals (three academic hospitals and six non-academic hospitals) in Sweden between 1 June 2020 and 17 May 2021.

### Protocol changes and rationale
The trial was designed in the beginning of the COVID-19 pandemic. After trial initiation, treatments for, and hospitalisation rates of, patients with COVID-19 changed rapidly. Therefore, we made protocol changes (described in detail in the online supplemental appendix) and the trial was stopped early.

In brief, we increased the number of study centres, removed the upper age limit (≤85 years) for patient inclusion, changed the inclusion criteria from ≤48 hours since hospital admission to ≤48 hours from initiation of oxygen therapy and allowed for patients to be included on the basis of a positive antigen test for SARS-CoV-2. All changes were approved by the Data Monitoring Committee, Ethical Review Authority and the Swedish Medical Products Agency and implemented from December 2020.

In June 2021, 99 patients had been included in the study, a large and increasing proportion of the adult Swedish population had received COVID-19 vaccination and hospitalisations for COVID-19 had dropped substantially. We determined that it was unlikely that the intended sample size would be reached and asked the Data Monitoring Committee to convene for a meeting. Following the recommendation of the Data Monitoring Committee, the study was terminated for futility to meet the targeted enrolment.

### Participants
Based on observations from patients with COVID-19 treated at the study centres, we expected that 85% of the standard care group would survive and terminate oxygen therapy within 30 days (median 8 days). We considered a 25% (2 days) reduction in the duration of oxygen therapy to be a clinically meaningful effect. We estimated that such an effect could be detected with α of 0.05, and 80% power if 446 participants (223 in each group) were enrolled.

Participants were eligible for inclusion if, they (1) were aged ≥18 years, (2) had a PCR confirmed SARS-CoV-2 infection or a positive SARS-CoV-2 antigen test from the upper respiratory tract, (3) were hospitalised at any of the study hospitals and (4) were receiving oxygen therapy, initiated within 48 hours before inclusion. Key exclusion criteria were ongoing treatment with inhaled or oral corticosteroids (previous use was accepted), oxygen therapy with >8 L oxygen/min or >50% oxygen on nasal high-flow cannula, and ongoing or expected intensive care or palliative care (online supplemental appendix).

### Randomisation
Patients were randomised 1:1 in blocks of 8, stratified by sex and hospital to receive ciclesonide or standard care. The randomisation sequence was prepared by a statistician not involved in the trial. Treatment allocation was provided through a web-based interface. The participants and the physicians treating them were unblinded to the treatment assignment.

### Intervention
The treatment was 320 µg of inhaled ciclesonide (80 µg per actuation, for a total of four actuations, or 160 µg per actuation, for a total of two actuations) two times a day (total daily dose 640 µg) for 14 days. Ciclesonide was administered using a spacer (L'espace, Nordic Infucare, Stockholm, Sweden). Participants randomised to ciclesonide received written instructions, including pictures, and practical instructions on how to use the inhalator and spacer; the first dose was taken under supervision. Ciclesonide was then prescribed in the participant's electronic medical record and each given dose during the hospitalisation was recorded. Participants discharged before day 14 were instructed to continue the treatment at home for a total treatment duration of 14 days. Participants randomised to standard care did not receive any intervention related to the study. Physicians treating the participants were not given any restrictions concerning treatments during the study period. Participants who had been discharged were contacted by telephone after day 30 for a follow-up interview.

### Outcomes
The primary outcome was duration of oxygen therapy (time to termination of oxygen therapy in days) up to 30

days from randomisation. Oxygen therapy was defined as terminated on the day after which the patient did not receive oxygen therapy during at least 48 hours, while being alive. This outcome corresponded to clinical improvement for patients receiving oxygen therapy according to the WHO clinical progression scale.[18]

The key secondary outcome was a composite of invasive mechanical ventilation and death up to 30 days after randomisation. Other secondary outcomes were each component of the key secondary outcome, admission to an intensive care unit, discharge from the hospital and dyspnoea in daily living at 30–35 days after randomisation as evaluated by the Modified Medical Research Council (mMRC) dyspnoea scale. The scale ranges from 0 to 4 with a higher score indicating more severe dyspnoea.[19 20]

Data on serious adverse events[21] were collected by review of electronic medical records. Information about non-serious adverse events associated with ciclesonide use (dryness of mouth, nausea and oral candidiasis) was reported using a paper-based reporting form which was filled in by the treating physician. Information about non-serious adverse events occurring after hospital discharge was collected during the follow-up interview.

### Data collection
Patient characteristics at baseline (comorbidities, comedications, clinical parameters) and study outcomes were obtained from electronic medical records. Investigators contacted participants after day 30 after randomisation to ask them about non-serious adverse events and dyspnoea in daily living (study outcome) at day 30–35 after randomisation.

### Statistical analysis
According to the pre-specified analysis plan in the study protocol, the analyses were performed by an investigator who had not been involved in the enrolment of participants and was blinded to treatment assignment. An intention-to-treat population was used. In the analysis of the duration of oxygen therapy, participants were followed from randomisation to termination of oxygen therapy, death or 30 days after randomisation. Kaplan-Meier cumulative incidence curves were generated to illustrate the cumulative incidence of termination of oxygen therapy in the ciclesonide and standard care groups. A Cox proportional hazard regression model, adjusted for study hospital (online supplemental appendix table 1), age (continuous variable) and sex was used to estimate HRs with 95% CI for time-to-event outcomes. Proportions and the absolute risk difference with 95% CI were presented for binary outcomes. In a per-protocol analysis of the primary outcome, participants assigned to ciclesonide were censored at the time of discontinuing treatment. The median mMRC score was compared using the Kruskal-Wallis test. A logistic regression model adjusted for study hospital, age and sex was used to compare the likelihood of reporting an mMRC score of 0 (dyspnoea only with strenuous exercise).

In an analysis that was not pre-specified, we additionally adjusted the primary outcome analysis for baseline variables, including days since symptom onset, C reactive protein and white cell count (as continuous variables), and diabetes mellitus, hypertension and hyperlipidaemia.

95% CIs of ratios not including 1 and 95% CIs for absolute risk differences not including 0 were considered statistically significant. Secondary outcome analyses were considered hypothesis-generating and no adjustment for multiple testing was made. Analyses were performed using Stata V.16.1 (StataCorp).

### Patient and public involvement
No patients were involved in setting the research question, nor in the design, conduct, or interpretation of the study.

### RESULTS
Of the 99 participants who underwent randomisation, 48 were assigned to receive ciclesonide and 51 to standard care (figure 1). One participant in the standard care group withdrew consent and was excluded from

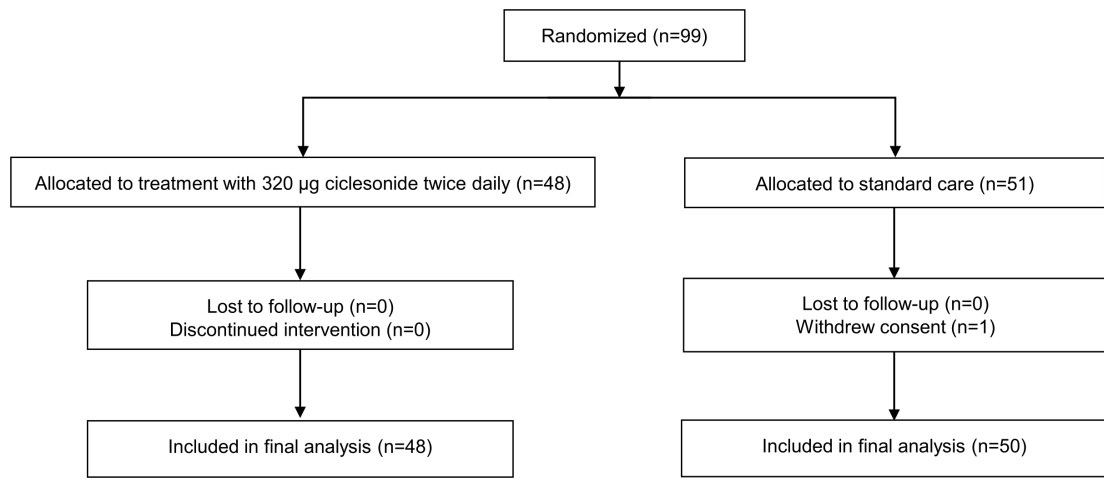

**Figure 1** Flow diagram for study participants.

**Table 1** Demographic and clinical characteristics of participants at study enrolment

| | Total (n=98) | Ciclesonide (n=48) | Standard care (n=50) |
|---|---|---|---|
| Age, median (IQR) | 59.5 (49–67) | 61 (49–67) | 59 (49–67) |
| Age <70 years, n (%) | 78 (80) | 37 (77) | 41 (82) |
| Men, n (%) | 67 (68) | 34 (71) | 33 (66) |
| Days since symptom onset, median (IQR) | 9 (8–11) | 9 (7.5–11.5) | 10 (8–11) |
| Days since symptom onset: <10 days, n (%) | 51 (52) | 27 (56) | 24 (48) |
| Body mass index, kg/m$^2$, median (IQR) | 29.7 (25.6–34.0) | 28.7 (25.4–34.0) | 30.6 (26.8–34.3) |
| Oxygen flow of oxygen therapy, L/min, median (IQR) | 2 (1–3) | 2 (1–3) | 2 (1–2) |
| Respiratory rate per minute, median (IQR) | 20 (18–24) | 20 (19–25) | 20 (18–23) |
| C reactive protein, mg/L, median (IQR) | 100 (56–142) | 103 (62–164) | 91.5 (45.5–124.5) |
| White cell count, 10$^9$/L, median (IQR) | 5.7 (4.5–7.0) | 5.3 (4.3–6.9) | 6.1 (4.9–7.0) |
| eGFR, mL/min/1.73 m$^2$, median (IQR) | 83 (70.5–90) | 81.5 (70–90) | 87 (73–90) |
| Coexisting conditions, n (%) | | | |
| Diabetes mellitus | 18 (18) | 8 (17) | 10 (20) |
| Hypertension* | 45 (46) | 22 (46) | 23 (46) |
| Hyperlipidaemia† | 27 (28) | 12 (25) | 15 (30) |
| Chronic obstructive lung disease | 3 (3) | 1 (2) | 2 (4) |
| Asthma | 8 (8) | 6 (13) | 2 (4) |
| Current smoker | 12 (12) | 6 (13) | 6 (12) |
| Ischaemic heart disease | 8 (8) | 2 (4) | 6 (12) |
| Heart failure | 3 (3) | 2 (4) | 1 (2) |
| Atrial fibrillation | 5 (5) | 3 (6) | 2 (4) |
| Cancer | 10 (10) | 5 (10) | 5 (10) |
| Chronic kidney disease | 9 (9) | 5 (10) | 4 (8) |

Missing values were: n=1 for days since symptom onset, n=20 for body mass index, n=1 for oxygen flow of oxygen therapy, n=1 for body temperature, n=1 for heart rate, n=3 for respiratory rate, n=3 for C reactive protein, n=7 for white cell count and n=22 for eGFR.
*Diagnosis of hypertension or use of antihypertensive drugs.
†Diagnosis of hyperlipidaemia or use of lipid lowering therapy.
eGFR, estimated glomerular filtration rate.

the analysis. Ninety-eight patients (48 in the ciclesonide group and 50 in the standard care group) were included in the final analysis. All participants assigned to ciclesonide received the treatment at least once. None of the participants were lost to follow-up. The median age of participants was 59.5 (IQR 49–67) years, 68% were men and the median duration of symptoms was 9 (IQR 8–11) days. There were no relevant between-group differences in demographic characteristics, laboratory test results or comorbidities at enrolment (table 1).

The results of primary and secondary outcome analyses are presented in table 2. Kaplan-Meier estimates of the median duration of oxygen therapy were 5.5 (IQR 3–9) days in the ciclesonide group and 4 (2–7) days in the standard care group (figure 2). The HR for termination of oxygen therapy during 30 days following randomisation, used to compare ciclesonide versus standard care, showed that ciclesonide treatment was not statistically significantly associated with the duration of oxygen therapy (0.73 (95% CI 0.47 to 1.11)).

The research question that we aimed to assess was whether inhaled ciclesonide, as compared with standard care, could reduce the time to clinical improvement (as indicated by duration of oxygen therapy). While the interpretation of statistically non-significant findings is a recurring and well-known subject of debate in the medical literature, it is generally not recommended to use a binary interpretation based on an arbitrary cut-off for statistical significance.[22–25] This is particularly important in this trial as it was terminated early and thereby underpowered to assess its primary outcome. However, it has been suggested that in trials with statistically non-significant findings, the 95% CIs should be used to rule in or rule out potential effect sizes of the intervention. In this study, we therefore assessed the largest benefit of ciclesonide that was compatible with the CI. Such a benefit was represented by the upper limit of the HR for time to termination of oxygen therapy (a higher HR indicates shorter duration of oxygen therapy for the ciclesonide group), that is, 1.11. We took the inverse of

**Table 2** Outcomes. All outcomes are recorded during 30 days following randomisation unless otherwise indicated

| | Ciclesonide | Standard care | Difference* |
|---|---|---|---|
| **Primary outcome** | | | |
| Duration of oxygen therapy, days, median (IQR) | 5.5 (3–9) | 4 (2–7) | 0.73 (0.47 to 1.11) |
| **Key secondary outcome** | | | |
| Death or invasive mechanical ventilation, n (%) | 3 (6) | 3 (6) | 0 (–9 to 10) |
| Time to death or invasive mechanical ventilation, days, median (IQR) | 2 (2–10) | 4 (2–7) | 0.90 (0.15 to 5.32) |
| **Secondary outcomes** | | | |
| Death, n (%) | 2 (4) | 1 (2) | – |
| Invasive mechanical ventilation, n (%) | 1 (2) | 3 (6) | – |
| Admission to an intensive care unit, n (%) | 4 (8) | 4 (8) | – |
| mMRC dyspnoea scale score at day 30–35, median (IQR)† | 3 (2–4) | 3 (2–4) | 0.97 |
| mMRC dyspnoea scale score 0 at day 30–35, n (%)† | 4 (9) | 7 (15) | 0.48 (0.11 to 2.04) |
| **Per protocol analysis‡** | | | |
| Duration of oxygen therapy, days, median (IQR) | 5 (3–9) | 4 (2–7) | 0.79 (0.51 to 1.23) |
| **Additionally adjusted analysis§** | | | |
| Duration of oxygen therapy, days, median (IQR) | 5 (3–9) | 4.5 (2–7) | 0.68 (0.43 to 1.09) |

*Differences are expressed as HRs (95% CI) estimated using a Cox proportional hazards model for time to event outcomes and as absolute risk difference (95% CI) in percent for outcomes of absolute risk. The comparison of the mMRC dyspnoea score was done using the Kruskal-Wallis test and the difference is expressed as a p value. The comparison of the likelihood of reporting an mMRC score of 0 was done using a logistic regression model and the difference is expressed as an OR (95% CI). Statistical testing for differences in proportions and time-to-event analyses were not performed for the secondary outcome events, including death, invasive mechanical ventilation and admission to an intensive care unit due to few events.
†Not including 1 participant in the standard care group and 2 participants in the ciclesonide group who died within 30 days of randomisation and 1 participant in the standard care group and 1 participant in the ciclesonide group with missing data on this outcome.
‡In the per-protocol analysis for duration of oxygen therapy, patients assigned to ciclesonide were censored at the time of discontinuing treatment.
§In addition to age, sex and study centre, this analysis of duration of oxygen therapy was adjusted for days since symptom onset, C reactive protein, white cell count (as continuous variables) and diabetes mellitus, hypertension and hyperlipidaemia (as categorical variables). The analyses included n=46 in the standard care group and n=45 in the ciclesonide group without missing data on any of the variables included in the model.
mMRC, modified Medical Research Council.

this HR (1/1.11=0.90) to calculate the relative reduction in duration of oxygen therapy that the HR was compatible with (ie, 1–0.90=10% relative reduction). We then multiplied this 10% relative reduction with the absolute duration of oxygen therapy in the standard care group

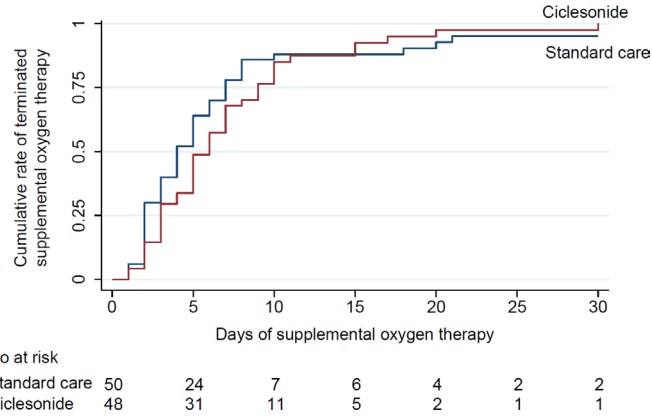

**Figure 2** Time to termination of oxygen therapy during 30 days after randomisation.

to calculate the corresponding absolute difference in duration of oxygen therapy (10%*4 days=0.4 days, which is <1 day). Given the pre-specified minimally clinically important difference of 2 days (which was used for the power calculation of the study), we deemed this best-case difference to be clinically irrelevant.

In the per-protocol analysis, the HR for termination of oxygen therapy during 30 days following randomisation was 0.79 (95% CI 0.51 to 1.23). In the additionally adjusted analysis (online supplemental appendix table 2), the HR for termination of oxygen therapy was 0.68 (95% CI 0.43 to 1.09) (table 2).

In total, three (6%) participants assigned to ciclesonide and three (6%) participants assigned to standard care experienced the key secondary outcome of mechanical invasive ventilation or death (absolute difference 0% (95% CI –10% to 9%; HR 0.90 (95% CI 0.15 to 5.32)). Median mMRC dyspnoea score at 30–35 days after randomisation was 3 (IQR 2–4) in both groups (p value for difference 0.97) (table 2).

**Table 3** Participants' treatments and adverse clinical events through day 30 after randomisation

| | Ciclesonide (n=48) | Standard care (n=50) |
|---|---|---|
| **Received treatment, n (%)** | | |
| Systemic corticosteroids | 26 (54) | 22 (44) |
| Remdesivir | 4 (8) | 5 (10) |
| Low-molecular-weight heparin | 45 (94) | 45 (90) |
| Oral anticoagulants | 32 (67) | 30 (60) |
| Vasopressors | 4 (8) | 3 (6) |
| Non-invasive mechanical ventilation | 8 (17) | 7 (14) |
| **Serious clinical events, n (%)** | | |
| Renal failure | 2 (4) | 3 (6) |
| Cardiac arrest | 1 (2) | 0 (0) |
| New onset atrial fibrillation | 0 (0) | 1 (2) |
| Pulmonary embolism | 4 (8) | 2 (4) |
| Other thromboembolic events | 0 (0) | 1 (2) |
| Sepsis | 3 (6) | 2 (4) |
| Other serious event | 1 (2) | 0 (0) |
| **Non-serious adverse events, n (%)** | | |
| Nausea | 6 (13) | 8 (16) |
| Dry mouth | 7 (15) | 11 (22) |
| Oral candidiasis | 2 (4) | 0 (0) |
| Other non-serious adverse event | 3 (6) | 1 (2) |

There were no apparent differences between the groups in treatments that participants received after randomisation (table 3); 26 (54%) of the participants assigned to ciclesonide and 22 (44%) of the participants in the standard care group received treatment with systemic corticosteroids after randomisation.

Few serious adverse clinical events occurred during the study. The most frequently reported adverse event was dry mouth (7 (15%) participants in the ciclesonide group and 11 (22%) participants in the standard care group). Two participants assigned to ciclesonide and 0 in the placebo group reported that they experienced oral candidiasis (table 3).

Some pre-specified analyses were not performed due to small sample size or low number of events. These included statistical testing of differences in proportions and time-to-event analyses for non-key secondary outcomes, including death, invasive mechanical ventilation and admission to an intensive care unit; the secondary outcome analyses of discharge from hospital; subgroup analyses, and the primary outcome analysis after exclusion of participants who received invasive mechanical ventilation or died.

## DISCUSSION

In this randomised open-label, controlled trial, including 98 patients hospitalised with COVID-19 with ongoing oxygen therapy, treatment with inhaled ciclesonide did not result in a statistically significant reduction in the duration of oxygen therapy, used as a measure of time to clinical improvement. The trial ruled out, with 0.95 confidence, treatments effects of ciclesonide corresponding to more than a 1 day reduction in duration of oxygen therapy.

While previous randomised controlled trials have assessed effects of inhaled corticosteroids, including budesonide[14 15] and ciclesonide,[16 17] in patients non-hospitalised with COVID-19, this is the first trial that includes hospitalised patients with more severe forms of the disease. In contrast to our hypothesis, the median duration of oxygen therapy was nominally longer among patients assigned to ciclesonide versus standard care (5.5 vs 4 days; HR for termination of oxygen therapy 0.73 (95% CI 0.47 to 1.11)). As such, the 95% CI indicates that,[24] even in the best case, ciclesonide may reduce the duration of oxygen therapy with only 10% (1-1/1.11; less than 1 day in our study) while it may in the worst case result in an over twofold increase. Thus, the results of this trial indicate that ciclesonide is unlikely to provide a clinically meaningful beneficial effect on the duration of oxygen therapy in patients hospitalised with COVID-19 receiving oxygen therapy.

To date, two randomised controlled trials of ciclesonide in patients non-hospitalised with COVID-19 have been presented. In the CONTAIN study,[16] which was terminated early due to slow recruitment, 215 non-hospitalised patients with a median of 3 days symptom duration were randomised to combination treatment with intranasal and inhaled ciclesonide or placebo. No statistically significant difference between the groups was observed for the primary endpoint, resolution of respiratory symptoms at day 7 after randomisation, which was reached by 40% of the patients in the treatment group versus 35% in the placebo group (adjusted risk difference of 5.5% (95% CI −7.8% to 18.8%).[16] Six (6%) patients assigned to ciclesonide versus 3 (3%) in the placebo group were hospitalised within 14 days; none died. In another clinical trial of ciclesonide, including 400 patients non-hospitalised with COVID-19,[17] randomisation to ciclesonid versus placebo did not result in a reduced time to alleviation of all COVID-19 related symptoms. However, in secondary outcome analyses, patients assigned to ciclesonide had fewer emergency department visits or hospital admissions for reasons related to COVID-19 (OR, 0.18, 95% CI, 0.04 to 0.85).

In addition, two randomised clinical trials of the inhaled corticosteroid budesonide in non-hospitalised patients with COVID-19 have been presented. The STOIC trial was an open-label trial comparing inhaled budesonide versus standard care in 146 patients with COVID-19 with mild symptoms.[14] Compared with standard care, budesonide treatment led to a statistically significant reduction in

COVID-19-related emergency department assessment and hospitalisation (difference in proportions 0.123 (95% CI 0.043 to 0.218)).[14] Furthermore, budesonide treatment was associated with 1 day shorter time to clinical recovery. The PRINCIPLE trial was another open-label trial that included 4700 primary care patients at high risk of developing severe COVID-19 (1073 randomised to budesonide treatment; 1988 to standard care; 1639 to other treatments).[15] Compared with standard care, randomisation to budesonide led to a shorter time to self-reported recovery (difference 2.94 days (95% Bayesian credible interval 1.19 to 5.12) and a reduced likelihood of hospital admission or death, although the results for the latter outcomes did not meet the superiority threshold.

Taken together, the previous studies indicate that inhaled corticosteroids might be useful for preventing deterioration of COVID-19 in patients non-hospitalised with mild symptoms. It is possible that the low likelihood of benefit associated with ciclesonide treatment observed in our study reflects the more severe pulmonary inflammation in our study population, as indicated by the need for hospitalisation with oxygen therapy and a median symptom duration of 9 days: at such stages of disease progression, it could be speculated that pulmonary administration of corticosteroids may not suffice to confer benefit and that systemic treatment is needed. Accordingly, in the Recovery trial of patients hospitalised with COVID-19,[26] dexamethasone treatment reduced risk of death and the time to discharge from hospital, with these benefits primarily being observed among patients receiving oxygen therapy or invasive mechanical ventilation at baseline.

Similar to other clinical trials including patients with COVID-19,[15 26 27] we used a pragmatic, open-label design. With this design, we intended to assess the effect of adding ciclesonide to standard care, rather than to examine the effect of ciclesonide compared with placebo. The research question that our study aimed to answer was 'what is the effect of using ciclesonide as an addition to standard care as compared with standard care alone?' While this is a research question of relevance to clinical decision-making, the open-label design and the possible expectations of effect among both patients[28] and physicians might have affected the outcomes in our study, including when to terminate oxygen therapy. Another limitation of our study is that we were unable to recruit the intended number of patients due to the substantial decrease in patients hospitalised with COVID-19 in Sweden during 2021. Importantly, the study could not provide much information regarding the key secondary outcome of death or invasive mechanical intervention. Further research in patients hospitalised with COVID-19 is needed to determine the potential effect of ciclesonide treatment on these outcomes. Moreover, it is a possibility that effects of ciclesonide differ as compared with other inhaled corticosteroids (eg, budesonide). Patients were instructed to use ciclesonide without a spacer after discharge from the hospital; this may have affected drug delivery. Finally, results from the Recovery Trial were released 5 weeks after the initiation of our study and around half of the patients in both the ciclesonide group and the control group received systemic corticosteroids after randomisation. Further studies would be needed to assess the comparative effectiveness and safety of ciclesonide versus systemic corticosteroids.

## CONCLUSIONS

In this open-label randomised controlled trial in patients hospitalised with COVID-19 and receiving oxygen therapy, the findings indicated that treatment with ciclesonide versus standard care is unlikely to result in a clinically meaningful reduction in the duration of oxygen therapy.

**Author affiliations**
[1]Department of Medicine, Capio S:t Göran's Hospital, Stockholm, Sweden
[2]Functional Area of Emergency Medicine, Karolinska Institute, Stockholm, Sweden
[3]Clinical Epidemiology Division, Department of Medicine, Solna, Karolinska Institutet, Stockholm, Sweden
[4]Department of Physiology and Pharmacology, Karolinska Institutet, Stockholm, Sweden
[5]Centre for Clinical Research Västmanland, Uppsala University, Uppsala, Sweden
[6]Department of Clinical Sciences, Danderyd Hospital, Stockholm, Sweden
[7]School of Medical Science, Örebro University, Örebro, Sweden
[8]Department of Internal Medicine, Danderyd Hospital, Stockholm, Sweden
[9]Department of Infectious Diseases, Central Hospital Växjö, Vaxjo, Sweden
[10]Department of Medicine and Geriatrics, Karlskoga Hospital, Karlskoga, Sweden
[11]Karolinska University Hospital, Stockholm, Sweden
[12]Department of Endocrinology, Karolinska University Hospital, Stockholm, Sweden
[13]Department of Molecular Medicine and Surgery, Karolinska Institutet, Stockholm, Sweden
[14]Department of Infectious Diseases, Halland's Hospital Halmstad, Halmstad, Sweden
[15]Department of Infectious Diseases, Visby Hospital, Visby, Sweden
[16]Department of Infectious Diseases, Östersund Hospital, Ostersund, Sweden
[17]Department of Infectious Diseases, Karolinska University Hospital, Stockholm, Sweden
[18]Department of Medicine Huddinge H7, Karolinska Institutet, Karolinska University Hospital, Stockholm, Sweden

**Acknowledgements** We would like to thank the following individuals that did not qualify for authorship but contributed to the study: Dr Oscar Bakhouch (Skaraborg Hospital), Dr Eva-Marie Boman and Dr Anders Lundqvist (Southern Älvsborg Hospital).

**Contributors** DB, PU, PT, OB and DPA conceived the study and were responsible for the methods. OB and DPA were responsible for the study conduct. DB, PU, OB and DPA were responsible for the financing. PT validated the data. PU performed the main analysis. DPA, PU and PT wrote the original draft of the manuscript. All authors wrote, reviewed, and edited the manuscript. OB and DPA supervised the study. DB, DPA, PU, PT and OB were responsible for administration of the project. DPA and OB are the guarantors. DB, PT, AK, EW, SA, SW, OE, AN, AE, JG, JE-K, SB, CU, ML, BJ, JLj, JLi, JH, OB and DPA enrolled participants in the study. The corresponding author attests that all listed authors meet authorship criteria and that no others meeting the criteria have been omitted.

**Funding** This study was funded by the Swedish Heart and Lung Foundation (Number 20200421), The Axel and Margaret Ax:son Johnson Foundation (N/A), CIMED (N/A), and Strategic Research Program at Karolinska Institutet (Number 961507), the Stockholm County Council (Number: 954970, 963296, 962029), and the Västmanland County Council (Grant nr LTV-938409). PU was supported by grants from the Strategic Research Program in Epidemiology at Karolinska Institutet (N/A), and a Faculty Funded Career Position at Karolinska Institutet (N/A). The funders had no role in the study design, conduct, collection, management, analysis, interpretation of data, writing or reviewing the manuscript or decision to submit

the manuscript for publication. The study drug was donated by COVIS Pharma but COVIS pharma did not participate in any other part of the study.

**Competing interests** This study received non-financial support from COVIS Pharma (study drug donation). The authors have no conflict of interest to disclose.

**Patient and public involvement** Patients and/or the public were not involved in the design, or conduct, or reporting, or dissemination plans of this research.

**Patient consent for publication** Not applicable.

**Ethics approval** This study involves human participants and was approved. The study was approved by the Swedish Ethical Review Authority (Ethics committee number 2020-02183) and the Swedish Medical Products Agency (Eudra-CT number 2020-001928-34) and registered at clinicaltrials.gov. Participants gave informed consent to participate in the study before taking part.

**Provenance and peer review** Not commissioned; externally peer reviewed.

**Data availability statement** Data are available upon reasonable request.

**ORCID iDs**
Per Tornhammar http://orcid.org/0000-0003-1043-1894
Eli Westerlund http://orcid.org/0000-0001-5453-1796
Daniel Peter Andersson http://orcid.org/0000-0003-4655-4837

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
