## [Reviewer comments · BMJ Open]

ARTICLE DETAILS

TITLE (PROVISIONAL)	Inhaled Ciclesonide in Adults Hospitalized with Covid-19: a Randomized Controlled Open-label Trial (HALT Covid-19)
AUTHORS	Brodin, Daniel; Tornhammar, Per; Ueda, Peter; Krifors, Anders; Westerlund, Eli; Athlin, Simon; Wojt, Sandra; Elvstam, Olof; Neumann, Anca; Elshani, Arsim; Giesecke, Julia; Edvardsson-Källkvist, Jens; Bunpuckdee, Sayam; Unge, Christian; Larsson, Martin; Johansson, Björn; Ljungberg, Johan; Lindell, Jonas; Hansson, Johan; Blennow, Ola; Andersson, Daniel Peter

VERSION 1 – REVIEW

REVIEWER	Mroueh, Salman American University of Beirut, Pediatrics and Adolescent Medicine
REVIEW RETURNED	11-Jul-2022

GENERAL COMMENTS	The study is well designed and the manuscript well written. The authors discuss the limitations of the study, the major one being the inability to enroll patients. I suspect this is not an uncommon occurrence in studies addressing Covid, as the characteristics of the patient population kept on changing with the advent of the vaccines and various treatment modalities, as well as the emergence of new variants of the virus.
--

REVIEWER	Crook, Joanne King's College Hospital NHS Foundation Trust, Pharmacy
REVIEW RETURNED	12-Jul-2022

GENERAL COMMENTS	Useful research question to look at effect of inhaled ciclesonide in adults with COVID-19 and well written Abstract- no mention of route of steroid (i.e inhaled) Intro - rationale described (and granted pre recovery) but what are benefits of inhaled versus oral steroids in this group requiring oxygen Method - Change of administration route on discharge to be without spacer - this could effect % delivery drug, especially if patients struggle with technique (?>85 yrs especially ? suitable) Explained to reduce wastage however can have a significant difference to drug delivery Inclusion - unclear as to when other inhaled/oral steroids were included and then sub analysis of this group -discusses as limitation and patients are evenly matched but changes research question as assume would have effect on outcome Some patients may have been receiving inhaled steroids prior to the trial? COPD or Asthmatics? Results - low recruitment numbers explained due to rapid changes in covid therapies, reported patients were mainly > 50years with
--

	comorbidities, who may have higher risk of adverse outcomes to covid compared to a younger cohort. Trial stopped early therefore underpowered to meet recruitment targets and statistical analysis
--	--

REVIEWER	Melikhov, Oleg Institute of Clinical Research
REVIEW RETURNED	25-Sep-2022

GENERAL COMMENTS	Using of interquartile range (instead of range) most suitable to measure/ present of variability for small skewed samples, when one tail is longer than the other. Probably this is a case but that couldn't be confirmed without data set. “(hazard ratio (HR) for termination of oxygen therapy 0.73 (95% CI 0.47-1.11)”: it is advisable to add more information about “termination”, for instance, “by day 14th”. Looks like “termination of oxygen therapy” occurred in all patients in both groups, sooner or later. “Median duration of oxygen therapy was 5.5 days in the ciclesonide group and 4 days in the standard care group”. Looks like oxygen therapy continued longer in ciclesonide group. The same time: “a treatment effect of ciclesonide corresponding to more than a one-day reduction in duration of oxygen therapy”. Please explain. “Being compatible with a 10% relative reduction in oxygen therapy duration, corresponding to a <1-day absolute reduction” also should be checked or explained.
--

REVIEWER	Megna, Rosario Istituto di Biostrutture e Bioimmagini Consiglio Nazionale delle Ricerche
REVIEW RETURNED	30-Sep-2022

GENERAL COMMENTS	The statistical analysis reported in the manuscript entitled “Inhaled Ciclesonide in Adults Hospitalized with Covid-19: a Randomized Controlled Open-label Trial (HALT Covid-19)” should be improved. The main criticism regards confidence intervals. The appropriate use of the confidence interval for samples of small size (i.e., approximately less than 30) requires a symmetric distribution of data (i.e., normal or quasi-normal distribution). Therefore, the authors should remove the confidence intervals in Table 2, where this condition is not verified. In order to obtain a comparison between groups of small size, the authors can use the Fisher exact test or Mann-Whitney U test, as appropriate. On the other hand, to avoid a low power of the Mann-Whitney U test, they should also use it with a total sample size approximately greater than 7. Consequently, the analysis of sub-groups that don't satisfy all required conditions should be removed. The authors should also report the comparison between Ciclesonide and Standard care groups in Table 1 and Table 3. In this way, readers can obtain an immediate comparison between sub-groups.
---

VERSION 1 – AUTHOR RESPONSE

Reviewer: 1

Dr. Salman Mroueh, American University of Beirut

Comments to the Author:

1. The study is well designed and the manuscript well written. The authors discuss the limitations of the study, the major one being the inability to enroll patients. I suspect this is not an uncommon occurrence in studies addressing Covid, as the characteristics of the patient population kept on changing with the advent of the vaccines and various treatment modalities, as well as the emergence of new variants of the virus.

Thank you.

Reviewer: 2

Mrs. Joanne Crook, King's College Hospital NHS Foundation Trust

Comments to the Author:

1. Useful research question to look at effect of inhaled ciclesonide in adults with COVID-19 and well written

Thank you.

3. Abstract- no mention of route of steroid (i.e inhaled)

This has now been clarified:

“Objective: To assess the efficacy of inhaled ciclesonide in reducing the duration of oxygen therapy (an indicator of time to clinical improvement) among adults hospitalized with Covid-19. “

“Intervention: Inhaled ciclesonide 320 µg twice daily for 14 days versus standard care.”

4. Intro - rationale described (and granted pre recovery) but what are benefits of inhaled versus oral steroids in this group requiring oxygen

We have now added an explanation regarding this issue:

“Moreover, potentially protective and preventive effects of inhaled corticosteroids for ARDS is supported by animals models of lung injury,(5-8) and in vitro studies,(9) and it has been speculated that local administration of the drug in the lung may maximize therapeutic benefits with fewer systemic side effects, as compared with systemic steroids.(Festic E, et al. Randomized Clinical Trial of a Combination of an Inhaled Corticosteroid and Beta Agonist in Patients at Risk of Developing the Acute Respiratory Distress Syndrome. Crit Care Med 2017; 45:798-805.)”

5. Method - Change of administration route on discharge to be without spacer - this could effect % delivery drug, especially if patients struggle with technique (?>85 yrs especially ? suitable) Explained to reduce wastage however can have a significant difference to drug delivery

We agree that the pragmatic approach of letting the patient use of the drug without a spacer after discharge may have affected drug delivery. We have now added this as a limitation in the limitations section of the Discussion:

“Patients were instructed to use ciclesonide without a spacer after discharge from the hospital; this may have affected drug delivery.”

6. Inclusion - unclear as to when other inhaled/oral steroids were included and then sub analysis of this group -discusses as limitation and patients are evenly matched but changes research question as

assume would have effect on outcome

We are not entirely sure about what is referred to in this comment. In the manuscript it is stated that ongoing treatment with inhaled or oral corticosteroids was an exclusion criterion (no revision):

“Key exclusion criteria were ongoing treatment with inhaled or oral corticosteroids,…”

We did not change the research question. In the manuscript it is stated that patients could be treated with systemic corticosteroids after randomization and described the number of treated subjects is described in Table 3 (no revision).

In the Discussion, we have mentioned that this was not a comparative effectiveness trial of ciclesonide vs systemic corticosteroids (no revision).

7. Some patients may have been receiving inhaled steroids prior to the trial? COPD or Asthmatics?

This is correct. Previous use of inhaled corticosteroids was not an exclusion criterion. We have now clarified this:

“Key exclusion criteria were ongoing treatment with inhaled or oral corticosteroids (previous use was accepted),…”

We have presented the number of patients in each group who had a previous diagnosis of COPD and asthma respectively (Table 1; no revision).

8. Results - low recruitment numbers explained due to rapid changes in covid therapies, reported patients were mainly > 50years with comorbidities, who may have higher risk of adverse outcomes to covid compared to a younger cohort. Trial stopped early therefore underpowered to meet recruitment targets and statistical analysis

This is correct and has been mentioned in the manuscript (no revision).

Reviewer: 3

Dr. Oleg Melikhov, Institute of Clinical Research

Comments to the Author:

1. Using of interquartile range (instead of range) most suitable to measure/ present of variability for small skewed samples, when one tail is longer than the other. Probably this is a case but that couldn't be confirmed without data set.

Thank you for the comments on our manuscript.

Given that the sample size was relatively small and many of the variables were not normally distributed (as represented by the median and IQR values), we chose to present descriptive data using median (IQR) rather than mean (SD).

2. “(hazard ratio (HR) for termination of oxygen therapy 0.73 (95% CI 0.47-1.11)”: it is advisable to add more information about “termination”, for instance, “by day 14th”. Looks like “termination of oxygen therapy” occurred in all patients in both groups, sooner or later.

We have now clarified that time to termination of oxygen therapy (duration of oxygen therapy) was only measured during 30 days after randomization. The Kaplan-Meier curves in Figure 2 show the proportion of patients who had their oxygen therapy terminated at different points in time after randomization.

Results:

“The HR for termination of oxygen therapy during 30 days following randomization, used to compare ciclesonide vs standard care,…”

“the per-protocol analysis, the HR for termination of oxygen therapy during 30 days following randomization was…”

Table 2 Outcomes. All outcomes are recorded during 30 days following randomization unless otherwise indicated.

“Figure 2 Time to termination of oxygen therapy during 30 days after randomization.”

3. “Median duration of oxygen therapy was 5.5 days in the ciclesonide group and 4 days in the standard care group”. Looks like oxygen therapy continued longer in ciclesonide group. The same time: “a treatment effect of ciclesonide corresponding to more than a one-day reduction in duration of oxygen therapy”. Please explain. “Being compatible with a 10% relative reduction in oxygen therapy duration, corresponding to a <1-day absolute reduction” also should be checked or explained.

Thank you for the opportunity to clarify. This point involves two comment concerning two related issues: a) the nominal difference in duration of oxygen therapy between the two groups; and b) how to interpret statistically non-significant findings: We discuss each issue below.

The nominal difference in duration of oxygen therapy between the two groups

The median duration of oxygen therapy was longer among patients receiving ciclesonide as compared with those receiving standard care (5.5 days vs 4 days). This finding could indicate that ciclesonide increases the duration of oxygen therapy (i.e., leads to harm). Accordingly, the HR for termination of oxygen therapy (with a lower HR indicating longer duration of oxygen therapy for the ciclesonide group) was (0.73 (95% CI 0.47 to 1.11), i.e. a statistically non-significant difference in time to termination of oxygen therapy between the groups, although the point estimate indicated that this time may be longer for the ciclesonide group.

Given that the study was terminated early and suffered from a small sample size, we would like to refrain from overinterpreting the point estimate and the non-significant results. This is also generally advised against according to established standards for reporting findings from clinical trials.

How to interpret statistically non-significant findings

Nonetheless, how to interpret statistically non-significant findings is a recurring and well-known problem in frequentist statistics. To simply use a binary interpretation based on an arbitrary cut-off for statistical significance is a heavily criticized approach.^{1–3}

Rather, the advice from medical statisticians is to interpret the point estimate and the confidence intervals to rule in and out possible effect sizes of the intervention. This would be especially important in underpowered and early terminated studies like ours.

The research question that we aimed to assess was whether inhaled ciclesonide, as compared with standard care, could reduce the time to clinical improvement (as indicated by duration of oxygen therapy). As such, when interpreting the 95% CI, we focused on the limit of the confidence interval indicating the maximum benefit of ciclesonide that the CI was compatible with (according to advice presented elsewhere^{1–3}).

The best case regarding the effect of ciclesonide that our data was compatible with was represented by the upper limit of the 95% CI for the HR (in the present study a higher HR indicates shorter duration of oxygen therapy for the ciclesonide group), namely 1.11, or an 11% relative increase in the risk (chance) of termination of oxygen therapy. We took the inverse of this HR ($1/1.11 = 0.90$) to calculate the relative reduction in duration of oxygen therapy that the HR was compatible with (i.e., $1 - 0.90 = 10\%$ relative reduction). We then multiplied this 10% relative reduction with the absolute duration of oxygen therapy in the standard care group to calculate the corresponding absolute difference in duration of oxygen therapy ($10\% * 4 \text{ days} = 0.4 \text{ days}$, which is <1 day). This is how we arrived at the estimate that this trial ruled out an absolute benefit of ciclesonide corresponding to a <1 day absolute reduction in the duration of oxygen therapy.

As the trial question pertained to potential benefit of ciclesonide, we did not assess the corresponding

worst case scenario (harm) compatible with the lower limit of the CI.

We agree that this calculation need to be explained in more detail and have now added an explanation in the Online Appendix. Moreover, the calculation was done post-hoc and was not pre-specified in the study protocol, which needs to be specified. We have now revised the manuscript to clarify:

Abstract:

“Median (IQR) duration of oxygen therapy was 5.5 (3-9) days in the ciclesonide group and 4 (2-7) days in the standard care group (hazard ratio (HR) for termination of oxygen therapy 0.73 (95% CI 0.47-1.11), with the upper 95% CI being compatible with a 10% relative reduction in oxygen therapy duration, corresponding to a <1-day absolute reduction in a post-hoc calculation).”

Results:

“The upper limit of the 95% CI was compatible with a maximum relative reduction(22) in duration of oxygen therapy of 10% (1-1/1.11) with ciclesonide, which, in a post-hoc calculation described in the Online Appendix, corresponded to a <1 day absolute reduction.”

Online Appendix

“Post-hoc calculation for interpretation of the 95% confidence interval in the primary outcome analysis
The research question that we aimed to assess was whether inhaled ciclesonide, as compared with standard care, could reduce the time to clinical improvement (as indicated by duration of oxygen therapy). While the interpretation of statistically non-significant findings is a recurring and well-known subject of debate in the medical literature, it is generally not recommended to use a binary interpretation based on an arbitrary cut-off for statistical significance.^{1–3} This is particularly important in this trial as it was terminated early and thereby underpowered to assess its primary outcome. However, it has been suggested that in trials with statistically non-significant findings, the 95% CIs should be used to rule in or rule out potential effect sizes of the intervention. In this study, we therefore assessed the largest benefit of ciclesonide that was compatible with the confidence interval. Such a benefit was represented by the upper limit of the HR for time to termination of oxygen therapy (a higher HR indicates shorter duration of oxygen therapy for the ciclesonide group), i.e., 1.11. We took the inverse of this HR ($1/1.11 = 0.90$) to calculate the relative reduction in duration of oxygen therapy that the HR was compatible with (i.e., $1 - 0.90 = 10\%$ relative reduction). We then multiplied this 10% relative reduction with the absolute duration of oxygen therapy in the standard care group to calculate the corresponding absolute difference in duration of oxygen therapy ($10\% * 4 \text{ days} = 0.4 \text{ days}$, which is <1 day). Given the pre-specified minimally clinically important difference of 2 days (which was used for the power calculation of the study), we deemed this best-case difference to be clinically irrelevant.”

Reviewer: 4

Dr. Rosario Megna, Istituto di Biostrutture e Bioimmagini Consiglio Nazionale delle Ricerche

Comments to the Author:

1. The statistical analysis reported in the manuscript entitled “Inhaled Ciclesonide in Adults Hospitalized with Covid-19: a Randomized Controlled Open-label Trial (HALT Covid-19)” should be improved.

The main criticism regards confidence intervals. The appropriate use of the confidence interval for samples of small size (i.e., approximatively less than 30) requires a symmetric distribution of data (i.e., normal or quasi-normal distribution). Therefore, the authors should remove the confidence intervals in Table 2, where this condition is not verified.

In order to obtain a comparison between groups of small size, the authors can use the Fisher exact

test or Mann-Whitney U test, as appropriate. On the other hand, to avoid a low power of the Mann-Whitney U test, they should also use it with a total sample size approximately greater than 7. Consequently, the analysis of sub-groups that don't satisfy all required conditions should be removed.

Thank you for this suggestion. Of the analyses presented in Table 2, only the subgroup analysis of those aged ≥ 70 years included less than 30 patients (sample size is specified in the table footnote; please note that the numbers presented in Table 2 do not represent sample size). We have now revised accordingly:

Table 2

Age group: ≥ 70 years

Duration of oxygen therapy, median (IQR) days 9 (5, 10) 6 (5, 8) 0.37 ($p=0.266d$)

d. 95% CI were not calculated due to low sample size. The p-value is calculated using the Mann-Whitney U Test for duration of oxygen therapy after exclusion of 1 patient in the standard care group and 2 patients in the ciclesonide group who died or received invasive mechanical intervention during 30 days after randomization. The use of the Mann-Whitney U Test was a post-hoc decision and the analysis could not be adjusted for age, sex and study hospital.

2. The authors should also report the comparison between Ciclesonide and Standard care groups in Table 1 and Table 3. In this way, readers can obtain an immediate comparison between sub-groups. In Table 1, we have now presented the n (%) of the subgroups used in the subgroup analyses (the number of study participants in each group is also presented in the footnote under Table 2). Now all variables used to create subgroups for subgroup analyses have been presented in Table 1.

Table 1 Demographic and clinical characteristics of participants at study enrolment.

Total (n=98) Ciclesonide (n=48) Standard care (n=50)

Age, median (IQR) 59.5 (49, 67) 61 (49, 67) 59 (49, 67)

Age < 70 years, n (%) 78 (80) 37 (77) 41 (82)

Men, n (%) 67 (68) 34 (71) 33 (66)

Days since symptom onset, median (IQR) 9 (8, 11) 9 (7.5, 11.5) 10 (8, 11)

Days since symptom onset: < 10 days, n (%) 51 (52) 27 (56) 24 (48)

Body mass index in kg/m², median (IQR) 29.7 (25.6, 34.0) 28.7 (25.4, 34.0) 30.6 (26.8, 34.3)

Oxygen flow of oxygen therapy in L/min, median (IQR) 2 (1, 3) 2 (1, 3) 2 (1, 2)

Respiratory rate per minute, median (IQR) 20 (18, 24) 20 (19, 25) 20 (18, 23)

...

We apologize if we have not fully understood the comment. If the comment refers to a suggestion of presenting the primary outcome analysis in all three tables, we would prefer not to do so as it is not an established way of presenting clinical trial results. Moreover, we want to emphasize that both Table 1 and Table 3 presents data for each group separately. If the comment refers to statistical testing of group differences (e.g., by t-test or Chi-squared test), we would like to refrain from it as it is not appropriate to use frequentist statistics testing to look for spurious (statistically significant) group differences when the intervention is assigned through randomization.

Reviewer: 1

Competing interests of Reviewer: None

Reviewer: 2

Competing interests of Reviewer: non

Reviewer: 3

Competing interests of Reviewer: No competing interests

Reviewer: 4

Competing interests of Reviewer: None

VERSION 2 – REVIEW

REVIEWER	Crook, Joanne King's College Hospital NHS Foundation Trust, Pharmacy
REVIEW RETURNED	14-Nov-2022

GENERAL COMMENTS	Thank you for taking on board my comments, I feel the manuscript is much easier to follow and understand. There is a small typo mistake in results -ciclesonide. Even though you have added an appendix and clarity to the explanation of HR post hoc review - I feel as a reader that it is over complex to have to read an appendix in order to analyse the results. Unfortunately, I can't advise a better way to do this but is worth considering
---

REVIEWER	Melikhov, Oleg Institute of Clinical Research
REVIEW RETURNED	09-Nov-2022

GENERAL COMMENTS	Thank you for giving me an opportunity to review your manuscript bmjopen-2022-064374 - "Inhaled Ciclesonide in Adults Hospitalized with Covid-19: a Randomized Controlled Openlabel Trial (HALT Covid-19)". I believe the results of the study will be of high scientific quality and will give a better understanding of the treatment of COVID-19. All my suggestions and comments from Review 1 are addressed properly. I recommend BMJ Open to accept the manuscript
--

REVIEWER	Megna, Rosario Istituto di Biostrutture e Bioimmagini Consiglio Nazionale delle Ricerche
REVIEW RETURNED	22-Nov-2022

GENERAL COMMENTS	Comment 1 Thanks for the clarification. I noted that the numbers in Table 2 do not represent the sample size. However, analyses were done between ciclesonide vs standard care group. Therefore, a normality test should be performed for each distribution with $n < 30$. For example, for the "Duration of oxygen therapy" in women, the authors should verify the normality of the distribution with "n=14 in the ciclesonide group" and "n=17 in the standard care group", and act accordingly. Please, also note that a sample with $n > 30$ does not guarantee a symmetric distribution (n value follows from the central limit theorem). Strictly speaking, even for this data, a normality test should be carried out.
--

	Comment 2 I apologize if I have not been clear; the comment refers to statistical testing. I might agree with the proposed approach, but with big numbers. Instead, with two groups of 48 and 50 patients, it would be desirable testing differences. For example, are the authors sure that there is no difference between the two groups with respect to the C-reactive protein variable (Table 1)? On the topic, institutions such as the European Medicines Agency [1] and the US Food and Drug Administration [2] advise of adjusting for baseline characteristics in the analysis of randomized clinical trials. Several authors are adopting this approach [3-6], and a practical guide has been recently published [7]. In the study under revision, the authors adjusted the model for study center, age and sex only for the primary outcome; clinical variables (such as C-reactive protein) were not included as covariates. However, a reader can evaluate differences for categorical variables (numbers are in tables), but he/she cannot evaluate differences for continuous variables. Concerning these variables, the authors should perform a test (t-test for normal distributions or Mann-Whitney U test otherwise) or give raw data in supplementary materials. Alternatively, they could adjust the model also for clinical variables. 1. European Medicines Agency. Guidelines on adjustment for baseline covariates in clinical trials. https://www.ema.europa.eu/en/documents/scientific-guideline/guideline-adjustment-baseline-covariates-clinical-trials_en.pdf 2. US Food and Drug Administration. Adjusting for covariates in randomized clinical trials for drugs and biological products. https://www.fda.gov/regulatory-information/search-fda-guidance-documents/adjusting-covariates-randomized-clinical-trials-drugs-and-biological-products 3. Localio AR, Berlin JA, Ten Have TR, Kimmel SE. Adjustments for center in multicenter studies: an overview. Ann Intern Med. 2001;135(2):112-123. doi:10.7326/0003-4819-135-2-200107170-00012 4. Kahan BC, Jairath V, Doré CJ, Morris TP. The risks and rewards of covariate adjustment in randomized trials: an assessment of 12 outcomes from 8 studies. Trials. 2014;15:139. doi:10.1186/1745-6215-15-139 5. Zampieri FG, Machado FR, Biondi RS, et al; BaSICS investigators and the BRICNet members. Effect of intravenous fluid treatment with a balanced solution vs 0.9%saline solution on mortality in critically ill patients: the BaSICS randomized clinical trial. JAMA. 2021;326(9):818-829. doi:10.1001/jama.2021.11684 6. Mistry EA, Yeatts SD, Khatri P, et al. National Institutes of Health Stroke Scale as an outcome in stroke research: value of ANCOVA over analysing change from baseline. Stroke. 2022;53(4):e150-e155. doi:10.1161/STROKEAHA.121.034859 7. Morris TP, Walker AS, Williamson EJ, White IR. Planning a method for covariate adjustment in individually randomised trials: a practical guide. Trials. 2022;23(1):328. doi:10.1186/s13063-022-06097-z
--	---

VERSION 2 – AUTHOR RESPONSE

Reviewer: 3

Dr. Oleg Melikhov, Institute of Clinical Research

Comments to the Author:

No.

Thank you for the comments on our manuscript.

Reviewer: 2

Mrs. Joanne Crook, King's College Hospital NHS Foundation Trust

Comments to the Author:

Thank you for taking on board my comments, I feel the manuscript is much easier to follow and understand. There is a small typo mistake in results -ciclesonide. Even though you have added an appendix and clarity to the explanation of HR post hoc review - I feel as a reader that it is over complex to have to read an appendix in order to analyse the results. Unfortunately, I can't advise a better way to do this but is worth considering

Thank you for the comments on our manuscript. We could not locate the typo.

We agree that the explanation of how the 95% CI of the primary outcome analysis is important to provide to the reader. We have therefore moved this explanation from the Appendix to the main paper (Results).

Reviewer: 4

Dr. Rosario Megna, Istituto di Biostrutture e Bioimmagini Consiglio Nazionale delle Ricerche

Comments to the Author:

Comment 1

Thanks for the clarification. I noted that the numbers in Table 2 do not represent the sample size.

However, analyses were done between ciclesonide vs standard care group. Therefore, a normality test should be performed for each distribution with $n < 30$. For example, for the "Duration of oxygen therapy" in women, the authors should verify the normality of the distribution with "n=14 in the ciclesonide group" and "n=17 in the standard care group", and act accordingly.

Please, also note that a sample with $n > 30$ does not guarantee a symmetric distribution (n value follows from the central limit theorem). Strictly speaking, even for this data, a normality test should be carried out.

Thank you for the comments on our manuscript and for highlighting the issues with small sample sizes in the subgroup analyses.

Even in a well powered clinical trial subgroup analyses are controversial. In frequentist statistics, when conducting several analyses, including subgroup analyses, spurious findings are inevitable, especially if sample size is small. Just like spurious positive findings may emerge in subgroup analyses, it can also happen that in certain subgroups, the point estimate may indicate no effect just by chance. A famous example of this phenomenon is the ISIS-2 study in which patients with myocardial infarction who were born under the Gemini and Libra star signs apparently experienced no benefit from aspirin vs placebo while for others, aspirin had a strongly beneficial effect.¹ It has therefore been recommended that subgroup analyses should be considered as exploratory and hypothesis-generating and that they are interpreted with greatest caution.^{2,3}

In our study, we did not reach the target sample size for the main analyses and as the reviewer points out, the even smaller sample sizes in the subgroup analyses introduce further uncertainty. Against

this background, we do not consider the subgroup analyses to provide any added value (if anything, they may introduce confusion). We have now removed all subgroup analyses from the paper.

Comment 2

I apologize if I have not been clear; the comment refers to statistical testing.

I might agree with the proposed approach, but with big numbers. Instead, with two groups of 48 and 50 patients, it would be desirable testing differences. For example, are the authors sure that there is no difference between the two groups with respect to the C-reactive protein variable (Table 1)?

On the topic, institutions such as the European Medicines Agency [1] and the US Food and Drug Administration [2] advise of adjusting for baseline characteristics in the analysis of randomized clinical trials. Several authors are adopting this approach [3-6], and a practical guide has been recently published [7].

In the study under revision, the authors adjusted the model for study center, age and sex only for the primary outcome; clinical variables (such as C-reactive protein) were not included as covariates.

However, a reader can evaluate differences for categorical variables (numbers are in tables), but he/she cannot evaluate differences for continuous variables. Concerning these variables, the authors should perform a test (t-test for normal distributions or Mann-Whitney U test otherwise) or give raw data in supplementary materials. Alternatively, they could adjust the model also for clinical variables.

1. European Medicines Agency. Guidelines on adjustment for baseline covariates in clinical trials. https://www.ema.europa.eu/en/documents/scientific-guideline/guideline-adjustment-baseline-covariates-clinical-trials_en.pdf
2. US Food and Drug Administration. Adjusting for covariates in randomized clinical trials for drugs and biological products. <https://www.fda.gov/regulatory-information/search-fda-guidance-documents/adjusting-covariates-randomized-clinical-trials-drugs-and-biological-products>
3. Localio AR, Berlin JA, Ten Have TR, Kimmel SE. Adjustments for center in multicenter studies: an overview. *Ann Intern Med.* 2001;135(2):112-123. doi:10.7326/0003-4819-135-2-200107170-00012
4. Kahan BC, Jairath V, Doré CJ, Morris TP. The risks and rewards of covariate adjustment in randomized trials: an assessment of 12 outcomes from 8 studies. *Trials.* 2014;15:139. doi:10.1186/1745-6215-15-139
5. Zampieri FG, Machado FR, Biondi RS, et al; BaSICS investigators and the BRICNet members. Effect of intravenous fluid treatment with a balanced solution vs 0.9% saline solution on mortality in critically ill patients: the BaSICS randomized clinical trial. *JAMA.* 2021;326(9):818-829. doi:10.1001/jama.2021.11684
6. Mistry EA, Yeatts SD, Khatri P, et al. National Institutes of Health Stroke Scale as an outcome in stroke research: value of ANCOVA over analysing change from baseline. *Stroke.* 2022;53(4):e150-e155. doi:10.1161/STROKEAHA.121.034859
7. Morris TP, Walker AS, Williamson EJ, White IR. Planning a method for covariate adjustment in individually randomised trials: a practical guide. *Trials.* 2022;23(1):328. doi:10.1186/s13063-022-06097-z

Thank you for highlighting this issue. Conducting statistical tests for potential group differences in a randomized controlled trial is not useful for reasons described, for example, in this piece.⁴ <https://www.nature.com/articles/s41393-018-0203-y>

We agree that adjusting for potential differences in baseline variables between the groups is useful, as such differences may affect the outcome, even if they have arisen by chance.

As suggested, we have therefore added a post-hoc analysis in which we adjust the model for main clinical variables and the most common comorbidities (in addition to age, sex and study center: days

since symptom onset, CRP, white blood count, diabetes mellitus, hypertension and hyperlipidaemia).

Methods:

In an analysis that was not pre-specified, we additionally adjusted the primary outcome analysis for baseline variables, including days since symptom onset, c-reactive protein and white blood count (as continuous variables), and diabetes mellitus, hypertension and hyperlipidemia.

Results:

“In the additionally adjusted analysis, the HR for termination of oxygen therapy was 0.68 (95% CI 0.43 to 1.09)”

Table 2 Outcomes. All outcomes are recorded during 30 days following randomization unless otherwise indicated.

Ciclesonide Standard care Difference^a

Primary outcome

Duration of oxygen therapy, median (IQR) days 5.5 (3, 9) 4 (2, 7) 0.73 (0.47 to 1.11)

Key secondary outcome

Death or invasive mechanical ventilation, n (%) 3 (6) 3 (6) 0 (-9 to 10)

Time to death or invasive mechanical ventilation, median (IQR) days 2 (2, 10) 4 (2, 7) 0.90 (0.15 to 5.32)

Secondary outcomes

Death, n (%) 2 (4) 1 (2) -

Invasive mechanical ventilation, n (%) 1 (2) 3 (6) -

Admission to an intensive care unit, n (%) 4 (8) 4 (8) -

mMRC dyspnea scale score at day 30-35, median (IQR)^b 3 (2, 4) 3 (2, 4) 0.97

mMRC dyspnea scale score 0 at day 30-35, n (%)^b 4 (9) 7 (15) 0.48 (0.11 to 2.04)

Per protocol analysis ^c.

Duration of oxygen therapy, median (IQR) days 5 (3, 9) 4 (2, 7) 0.79 (0.51 to 1.23)

Additionally adjusted analysis^d

Duration of oxygen therapy, median (IQR) days 5 (3, 9) 4.5 (2, 7) 0.68 (0.43 to 1.09)

a. Differences are expressed as hazard ratios (95% CI) estimated using a Cox proportional hazards model for time to event outcomes and as absolute risk difference (95% CI) in percent for outcomes of absolute risk. The comparison of the mMRC dyspnea score was done using the Kruskal-Wallis test and the difference is expressed as a p-value. The comparison of the likelihood of reporting a mMRC score of 0 was done using a logistic regression model and the difference is expressed as an odds ratio (95% CI). Statistical testing for differences in proportions and time-to-event analyses were not performed for the secondary outcome events, including death, invasive mechanical ventilation, and admission to an intensive care unit due to few events.

b. Not including 1 participant in the standard care group and 2 participants in the ciclesonide group who died within 30 days of randomization and 1 participant in the standard care group and 1 participant in the ciclesonide group with missing data on this outcome.

c.. In the per-protocol analysis for duration of oxygen therapy, patients assigned to ciclesonide were censored at the time of discontinuing treatment.

d. In addition to age, sex and study center, this analysis of duration of oxygen therapy was adjusted for days since symptom onset, c-reactive protein, white blood count (as continuous variables) and diabetes mellitus, hypertension and hyperlipidemia (as categorical variables). The analyses included n=46 in the standard care group and n=45 in the ciclesonide group without missing data on any of the variables included in the model.

Reviewer: 3

Competing interests of Reviewer: No competing interests.

Reviewer: 2

Competing interests of Reviewer: non

Reviewer: 4

Competing interests of Reviewer: None

References

1 Horton R. From star signs to trial guidelines. *Lancet* 2000; 355: 1033–4.

2 Sun X, Briel M, Walter SD, Guyatt GH. Is a subgroup effect believable? Updating criteria to evaluate the credibility of subgroup analyses. *BMJ* 2010; 340: 850–4.

3 Burke JF, Sussman JB, Kent DM, Hayward RA. Three simple rules to ensure reasonably credible subgroup analyses. *BMJ* 2015; 351. DOI:10.1136/BMJ.H5651.

4 Harvey LA. Statistical testing for baseline differences between randomised groups is not meaningful. *Spinal Cord* 2018 56:10 2018; 56: 919–919.

VERSION 3 – REVIEW

REVIEWER	Megna, Rosario Istituto di Biostrutture e Bioimmagini Consiglio Nazionale delle Ricerche
REVIEW RETURNED	24-Jan-2023
GENERAL COMMENTS	Thanks to the authors for their revisions. Now the manuscript is improved, and the research has greater statistical robustness. Please, put the complete multivariable logistic regression outcomes related to the adjusted models in the supplementary materials.

VERSION 3 – AUTHOR RESPONSE

Thank you for your constructive feedback. We have now included all the analyses in the supplementary material with reference from the manuscript.